# Association Study between the Polymorphisms of Matrix Metalloproteinase (MMP) Genes and Idiopathic Recurrent Pregnancy Loss

**DOI:** 10.3390/genes10050347

**Published:** 2019-05-07

**Authors:** Han Sung Park, Ki Han Ko, Jung Oh Kim, Hui Jeong An, Young Ran Kim, Ji Hyang Kim, Woo Sik Lee, Nam Keun Kim

**Affiliations:** 1Department of Biomedical Science, College of Life Science, CHA University, Seongnam 13488, Korea; hahnsung@naver.com (H.S.P.); fise123@naver.com (K.H.K.); jokim8505@gmail.com (J.O.K.); tody2209@naver.com (H.J.A.); 2Department of Obstetrics and Gynecology, CHA Bundang Medical Center, CHA University, Seongnam 13496, Korea; happyiran@cha.ac.kr; 3Fertility Center of CHA Bundang Medical Center, CHA University, Seongnam 13496, Korea; bin0902@chamc.co.kr; 4Fertility Center of CHA Gangnam Medical Center, CHA University, Seoul 06125, Korea

**Keywords:** recurrent pregnancy loss (RPL), matrix metalloproteinase (MMP), *MMP-8*, *MMP-27*, polymorphism

## Abstract

Recurrent pregnancy loss (RPL) refers to two or more consecutive pregnancy losses. It is estimated that fewer than 5% of women experience RPL. Matrix metalloproteinases (MMPs) are a family of proteolytic enzymes that play important roles in providing a safe and conducive environment for the stable development of the fetus. In this case-control study, we evaluated the associations between RPL and single nucleotide polymorphisms (SNPs) in *MMP-8* and *MMP-27*. We recruited 375 Korean women with a history of RPL and 240 ethnically-matched healthy parous controls, and we performed genotyping for the *MMP-8* rs2509013 C>T, *MMP-8* rs11225395 G>A, and *MMP-27* rs3809017 T>C polymorphisms. All SNPs were genotyped via the polymerase chain reaction–restriction fragment length polymorphism (PCR-RFLP) assay. In the genotype frequency analyses, the TT genotype of the *MMP-8* rs2509013 C>T (age-adjusted odds ratio, 0.415; 95% confidence interval, 0.257–0.671; *P* = 0.0003) and TC genotype of *MMP-27* rs3809017 T>C (age-adjusted odds ratio, 0.681; 95% confidence interval, 0.483–0.961; *P* = 0.029) were associated with decreased RPL susceptibility. Moreover, these trends were maintained in the haplotype and genotype combination analyses. Interestingly, amongst the RPL patients, higher levels of homocysteine (*P* = 0.042) and uric acid (*P* = 0.046) were associated with *MMP-27* rs3809017 T>C. In conclusion, the two polymorphisms of *MMP-8* and *MMP-27* were significantly associated with RPL risk, both individually and in combination. Therefore, these two polymorphisms are potential biomarkers for RPL susceptibility.

## 1. Introduction

Pregnancy loss, also referred to as spontaneous abortion, is defined as the loss of a clinical pregnancy prior to 20 weeks of gestational age. Recurrent pregnancy loss (RPL) refers to consecutive pregnancy losses; however, the European Society for Human Reproduction and Embryology and the Royal College of Obstetricians and Gynecologists have defined RPL as the loss of three or more consecutive pregnancies. More recently, the American Society for Reproductive Medicine broadened the definition of RPL to two or more consecutive pregnancy losses [1]. RPL is a significant and distressing clinical concern, since 2–5% of couples that hope to have a baby experience RPL. Various etiological factors are reported to contribute to RPL, including thrombosis, advanced maternal age, maternal anatomic anomalies, chromosomal abnormalities, endocrine system dysfunction, antiphospholipid syndrome, immunological problems, uterine anomalies, hereditary thrombophilia, and environmental factors [2]. However, the underlying causes related to 30–40% of RPL cases are still unclear and several studies investigating the associations between potential genetic factors and RPL are currently underway.

Matrix metalloproteinases (MMPs) are a family of proteolytic enzymes that play important roles in the degradation and remodeling of the extracellular matrix (ECM). During pregnancy, maternal bodies undergo dynamic changes that provide a safe and conducive environment for stable fetal development. Many of these anatomical changes occur in the uterus and in the endometrium, and they involve ECM remodeling by the MMPs, especially during embryo implantation, trophoblast invasion, and early placentation. MMP-8 is a neutrophil collagenase that preferentially degrades type I collagen, which normally provides tensile strength to the ECM of the cervix, uterus, and fetal membranes [3]. Additionally, the MMP-8 levels are regulated by neutrophil degranulation. MMP-27 belongs to the stromelysin group of MMPs, which includes MMP-3, MMP-10, and MMP-11. MMP-27 has a 51.6% homology with MMP-10, whilst little is known about its molecular functions [4]. The mRNA transcripts of *MMP-27* have been detected in most tissues during mouse development [5], and they are also abundant in rat bone and kidneys [6]. The levels of MMP-27 vary dynamically in the M2 macrophages of the human endometrium, based on the phase of the menstrual cycle [7]. Macrophages play key roles in a variety of processes that are required for a successful pregnancy, such as remodeling of the uterine connective tissues and blood vessels, as well as the regulation of trophoblast implantation [8,9].

Abnormal MMP expression and mutated MMPs are associated with various diseases, including diseases of the cardiovascular system, central nervous system, and reproductive system [10]. MMP-2 and MMP-9 are overexpressed in a human myocardial infarction, and the expression of MMP-1 and MMP-3 is associated with a protective effect in the development of atherosclerosis [10]. Additionally, a genetic variant in the promoter of human MMP-9 that leads to this increased expression was found to be associated with more severe atherosclerosis [11]. The levels of MMP-8 found in the cervical fluid of pregnant women are different according to term of pregnancy, where elevated amniotic fluid MMP-8 levels are associated with an increased risk of spontaneous preterm delivery. Some single nucleotide polymorphisms (SNPs) of *MMP-8* have been associated with osteonecrosis of the femoral head [12] and one of these SNPs (rs11225395) was associated with the susceptibility to severe preeclampsia [13]. The mRNAs of *MMP-27* were up-regulated in patients with abdominal aortic aneurysms [14] and they were down-regulated in the macrophages co-cultured with ovarian cancer cells [15]. Additionally, *MMP-8* and *MMP-27* were the most mutated *MMP* genes in melanomas [16]. Given the weight of evidence implicating SNPs in *MMP* genes in the susceptibility to various medical conditions, we investigated the potential associations between RPL and polymorphisms of: (1) *MMP-8,* that have been found to be associated with spontaneous preterm delivery (*MMP-8* rs2509013 C>T and *MMP-8* rs11225395 G>A), and (2) *MMP-27* gene, which is less well known (*MMP-27* rs3809017 T>C), using a polymerase chain reaction (PCR)–restriction fragment length polymorphism (RFLP) in a case-control study involving a population of Korean women.

## 2. Materials and Methods 

### 2.1. Study Population

We collected blood samples from 375 patients with idiopathic RPL (i.e., patients who had experienced at least two consecutive spontaneous abortions, each before 20 weeks of gestation) and 240 ethnically-matched control participants. Spontaneous abortions in the RPL patients were verified by analysis of the human chorionic gonadotropin levels, ultrasonography, and physical examination. No participant in this study had a history of smoking or alcohol use. Patients with pregnancy loss due to anatomic, hormonal, chromosomal, infectious, autoimmune, or thrombotic causes were excluded from the study group. To identify anatomic abnormalities in the RPL patients, we used sonography, hysterosalpingogram, hysteroscopy, computerized tomography, and magnetic resonance imaging. Hormonal causes included hyperprolactinemia, luteal insufficiency, and thyroid disease, and these causes were evaluated by measuring the levels of prolactin, thyroid-stimulating hormone, free T4, follicle-stimulating hormone, luteinizing hormone, and progesterone in the peripheral blood. Patients in whom the pregnancy loss was attributed to autoimmune causes were defined as patients diagnosed either with antiphospholipid syndrome or with lupus (diagnosed using lupus anticoagulant and anticardiolipin antibodies). Thrombotic causes were defined as thrombophilia and these were evaluated by deficiencies in protein C and protein S, and by the presence of the anti-β2 glycoprotein antibody. Chromosomal analyses of the blood samples were carried out using standard protocols as described in Reference [17]. All the participants were enrolled in the study at the Infertility Medical Center of the CHA Bundang Medical Center (Seongnam, Korea) between March 1999 and February 2012. The women in the control group included healthy women with regular menstrual cycles, normal karyotype of 46XX, a history of at least one naturally conceived pregnancy, and no history of pregnancy loss. The Institutional Review Board of CHA Bundang Medical Center (IRB number: BD2010-123D) approved the study and all the patients gave their written informed consent.

### 2.2. Genotyping

Blood samples were collected into ethylenediaminetetraacetic acid (EDTA)-coated blood collection tubes. The sample tubes were centrifuged for 15 min at 1000 × g to separate the plasma and buffy coat from the whole blood. The plasma was collected in 1.5 mL tubes and preserved in a deep-freezer at -80 ℃ until assessment of the baseline biochemical parameters. The patients’ buffy-coat samples were used for genomic DNA extraction using the G-DEX blood extraction kit (Intron, Seongnam, South Korea). Two SNPs of *MMP-8* (rs2509013 C>T and rs11225395 G>A) and a polymorphism of *MMP-27* (rs3809017 T>C) were determined by PCR-RFLP analysis using the isolated genomic DNA as a template, as in Reference [18]. The primer pairs for *MMP-8* rs2509013 C>T (forward: 5’ – CCT GGA AAG GCA CCT GAT ATG – 3’, and reverse: 5’ – CCT AGT AAA CAG GGC ATT GTG A – 3’), *MMP-8* rs11225395 G>A (forward: 5’ – TTC ACA TAG CCT TGG GAG G – 3’, and reverse: 5’ - TGG GAG ACT ACC ATG CAG ATC – 3’), and *MMP-27* rs3809017 T>C (forward: 5’ – ATT CTT TTC AGG GAT TCT GTA GAT T – 3’, and reverse: 5’ – TCT GGG TGG TGA CAC TCA TT – 3’) were designed to amplify each SNP, along with its flanking region. The conditions for PCR amplification were as follows: initial denaturation was performed at 95 °C for 15 min, followed by 35 cycles with denaturation at 95 °C for 30 sec, annealing for 30 sec, and an extension at 72 °C for 30 sec. A final extension was carried out at 72 °C for 5 min. The annealing temperatures for the *MMP-8* rs2509013 C>T, *MMP-8* rs11225395 G>A, and *MMP-27* rs3809017 T>C primer pairs were 62 °C, 58 °C, and 46 °C, respectively. The PCR products for the rs2509013, rs11225395, and rs3809017 were digested using *Bts*CI, *Bgl*II, and *Ase*I restriction enzymes (New England BioLabs, Ipswich, MA, USA), respectively, at 37 °C for 16 h.

### 2.3. Assessment of the Plasma PAI-1, Homocysteine, Folate, Total Cholesterol, Uric Acid, and Blood Coagulation Times

Isolated plasma was used in the assessment of a panel of biochemical parameters. The levels of PAI-1 were evaluated via a human SERPINE1/PAI-1 immunoassay (R&D Systems, Minneapolis, MN, USA). The levels of homocysteine were measured using the IMx fluorescence polarizing immunoassay (Abbott Laboratories, Abbott Park, IL, USA), and the levels of folate were measured using a radio-assay kit (ACS:180; Bayer, Tarrytown, NY, USA). Total cholesterol (T. chol) and uric acid levels were determined using commercially available enzymatic colorimetric tests (Roche Diagnostics GmbH, Mannheim, Germany). Platelet (PLT) counts were measured on a Sysmex XE2100 automated hematology analyzer (Sysmex Corporation, Kobe, Japan). Prothrombin time (PT) and activated partial thromboplastin time (aPTT) were measured using an automated photo-optical coagulometer ACL TOP (Mitsubishi Chemical Medience, Tokyo, Japan).

### 2.4. Estimation of the CD56+ Peripheral Natural Killer (NK) Cell Proportion

Peripheral blood mononuclear cells (PBMCs, 2.5×105) were stained at 4 °C in the dark for 30 min, and then washed twice in 2 mL phosphate-buffered saline (PBS) containing 1% bovine serum albumin and 0.01% sodium acid (FACS wash buffer). They were subsequently fixed in 200 μL of 1% paraformaldehyde (Sigma-Aldrich, St. Louis, MO, USA) prior to sorting as described in References [19,20]. 

Estimation of the CD56+ NK cell proportion in peripheral blood was performed using flow cytometry (FACSCalibur, BD Biosciences, San Jose, CA, USA). The flow cytometer was compensated by a single fluorochrome, and the data were analyzed using the CellQuest software (BD Biosciences). CD56-specific monoclonal antibody labelled with a fluorochrome (fluorescein isothiocyanate, phycoerythrin, peridinin-chlorophyll-protein complex, or allophycocyanin, BD Biosciences), and anti-NK cell receptor A-phycoerythrin (Immunotech, Beckman Coulter, Fullerton, CA, USA) were used for the NK cell detection. The PBMCs were gated based on the forward and side scatter to allow for the identification of lymphocytes. Positive and negative thresholds for the fluorescence signals were defined using isotype-specific negative controls. All the flow cytometric analyses were performed using FlowJo (Treestar, Inc. Ashland, OR, USA).

### 2.5. Statistical Analysis 

Differences in the genotype and haplotype frequencies between the RPL patients and the control groups were compared using multivariate logistic regression and the Fisher’s exact test, respectively. The odds ratio (OR), adjusted odds ratio (AOR), and 95% confidence intervals (CIs) were used to examine the association between *MMP* gene polymorphisms and RPL risk. The data are presented as mean ± SD (continuous variables) or percentages (categorical variables). The Hardy–Weinberg equilibrium (HWE) p-values of genotype frequencies were estimated using the Chi-square test to identify deviations from the HWE. The false-positive discovery rate (FDR) test was used to adjust multiple comparison tests and an FDR *p*-value < 0.05 was considered to be statistically significant as in Reference [21]. Differences in the plasma homocysteine, folate, PLT counts, NK cell, PT, uric acid, T. chol, and PAI-1 between different *MMP* genotypes were evaluated using the one-way analysis of variance (ANOVA). Statistical significance was accepted at *p* < 0.05. Analyses were performed using GraphPad Prism (v.4.0, GraphPad Software Inc. San Diego, CA, USA) and Medcalc (v. 12.7.1.0, Medcalc Software, Mariakerke, Belgium). The HAPSTAT program (v.3.0, www.bios.unc.edu/~lin/hapstat/) was used with a strong synergistic effect to estimate the frequency of polymorphic haplotypes and *p* < 0.05 was considered to be statistically significant.

## 3. Results

### 3.1. Study Population

We first evaluated if there were major differences in the clinical characteristics of patients with and without a history of RPL. The baseline clinical characteristics and biochemical profiles of RPL patients and the controls are presented as the mean ± standard deviation (SD) in Table 1. The mean ages of the RPL patients (mean ± SD, 33.19 ± 5.43) and the controls (mean ± SD, 33.21 ± 4.60) were not significantly different. Average gestational weeks of the RPL patients was 7.33 ± 1.85 (mean ± SD) and the proportion of less than 10 gestational weeks was 89.4%. The mean BMI of RPL patients (mean ± SD, 21.71 ± 3.45) and the controls (mean ± SD, 21.50 ± 3.89) were not different, and they were in the normal range. Although the RPL patients had differences in the levels of PLT count, aPTT, and PT compared to the controls, these parameters were still in the normal range.

### 3.2. Genotype Frequencies of the Polymorphisms in the MMP Genes

We investigated the differences in the genotype frequencies of the RPL patients and controls (Table 2). The age-adjusted odds ratios (AORs) were calculated using logistic regression analysis. In the patients with two or more pregnancy losses (RPL) versus the controls, the minor homozygous genotype (TT) frequency of *MMP-8* rs2509013 C>T (CC vs. TT: AOR, 0.415; 95% CI, 0.257–0.671; *p* = 0.0003), and the heterozygous genotype (TC) frequency of *MMP-27* rs3809017 T>C (TT vs. TC: AOR, 0.681; 95% CI, 0.483–0.961; *p* = 0.029) were significantly associated with the decreased risk of RPL. In patients with three or more pregnancy losses versus the controls, the TT genotype frequency of *MMP-8* rs2509013 C>T (CC vs. TT: AOR, 0.424; 95% CI, 0.219–0.822; *p* = 0.011) was significantly associated with the decreased risk of RPL. Additionally, the *MMP-8* rs2509013 TT genotype was associated with a reduced susceptibility to idiopathic RPL when compared to the CC and CC + CT genotypes, and this association remained statistically significant in the FDR test as well.

### 3.3. Haplotype Analysis

To evaluate whether the gene–gene interactions on the same chromosome had synergistic effects on decreasing the RPL risk, we performed haplotype analysis of the three SNPs (Table 3). In the haplotype analysis of three SNPs (*MMP-8* rs2509013 C>T/*MMP-8* rs11225395 G>A/*MMP-27* rs3809017 T>C), most haplotypes that had the T allele of *MMP-8* rs2509013 C>T or the C allele of *MMP-27* rs3809017 T>C, were associated with a decreased RPL risk compared to the reference haplotype (C-G-T). In the haplotype analysis of *MMP-8* rs2509013 C>T and *MMP-8* rs11225395 G>A, all haplotypes that included the T allele of *MMP-8* rs2509013 C>T were associated with a decreased RPL risk. Furthermore, the haplotype analyses revealed that some haplotypes that included the T allele of *MMP-8* rs2509013 C>T or the C allele of *MMP-27* rs3809017 T>C, were associated with a decreased RPL risk compared to the respective reference haplotypes. 

### 3.4. Genotype Combination Analysis 

Given the polygenic etiology of RPL, we reasoned that the analysis of combinations of different *MMP* polymorphisms might yield novel insights into the drivers and/or modifiers of RPL risk. Therefore, we conducted a genotype combination analysis involving the three polymorphisms of the *MMP* genes (Table 4). In the combination analysis of the *MMP-8* rs2509013 and rs11225395 genotypes, two combinations that included the TT genotype of *MMP-8* rs2509013 were associated with a decreased RPL risk. A combination analysis of the *MMP-8* rs2509013 and *MMP-27* rs3809017 genotypes revealed that almost all the combinations were associated with a decreased RPL risk. In the combination analysis of the *MMP-8* rs11225395 and *MMP-27* rs3809017 genotypes, only one combination that included the TC genotype of *MMP-27* rs3809017 showed an association with a decreased RPL risk.

### 3.5. Differences in the Clinical Parameters of RPL Patients according to Genotypes of the MMP Polymorphisms

Although previous studies have found that women with RPL show alterations in serum factors and biochemical parameters, none of these factors were found to reliably predict repeated miscarriages. We sought associations between the polymorphisms in *MMP-8* and *MMP-27* and the baseline biochemical parameters in the blood samples of RPL patients. In patients with two or more pregnancy losses, the *MMP-27* rs3809017 T>C polymorphism was associated with relatively higher levels of homocysteine (mean values for different genotypes: TT, 7.27 ± 2.31; TC, 6.63 ± 1.63; CC, 6.85 ± 1.98; *p* = 0.042) and uric acid (mean values for different genotypes: TT, 3.95 ± 0.88; TC, 3.62 ± 0.80; CC, 3.68 ± 0.65; *p* = 0.046). In the subgroup of patients with ≥3 pregnancy losses, the *MMP-27* rs3809017 T>C polymorphism was associated with higher levels of homocysteine (mean values for different genotypes: TT, 7.25 ± 1.78; TC, 6.24 ± 1.40; CC, 6.64 ± 1.61; *p* = 0.006) (Table 2 and Table 5).

## 4. Discussion

It is well established that several genetic, physiological, and even environmental factors underlie the complex etiology of RPL [22]. In this case-control study, we evaluated the potential associations between the SNPs in *MMP* genes (*MMP-8* rs2509013 C>T, *MMP-8* rs11225395 G>A, *MMP-27* rs3809017 T>C) and patients’ susceptibility to idiopathic RPL. Our results showed that *MMP-8* rs2509013 C>T and *MMP-27* rs3809017 T>C polymorphisms were significantly associated with decreased RPL susceptibility. Furthermore, our haplotype analyses showed that the allelic combinations, including the T allele of *MMP-8* rs2509013 and the C allele of *MMP-27* rs3809017, were associated with a decreased risk of RPL.

MMPs play a significant role in the degradation and rebuilding of the ECM. Numerous MMPs are highly expressed in many reproductive processes, including menstruation, ovulation, implantation, and angiogenesis [20,23]. MMPs can be regulated by reproductive hormones, which cause a dynamic alteration in the reproductive tissues. These pregnancy-assisting alterations provide a conducive environment for fertilization, implantation, and fetal growth. Therefore, MMPs are crucial to these reproductive processes, and the abnormal function or expression of MMPs may be critical for successful reproduction.

MMP-8 is also known as a polymorphonuclear neutrophils (PMNL) collagenase and the *MMP-8* gene is located on 11q22.2. MMP-8 is detectable in cervical fluid in almost all pregnant women in early and mid-pregnancy [24]. Moreover, MMP-8 has been reported to be present in amniotic fluid and fetal membranes [24]. Additionally, MMP-8 is used as one of the biomarkers of preterm delivery because high concentrations of MMP-8 in the cervical fluid are associated with spontaneous preterm delivery [3]. Therefore, MMP-8 may be a crucial player in RPL, which may be caused by similar mechanisms. *MMP-8* polymorphisms, rs2509013 C>T and rs11225395 G>A, are located in an intron between the 8th and 9th exons and in the promoter region at the 934 bp upstream of start codon of *MMP-8*, respectively. The intron variant rs2509013 has been studied in the context of certain diseases, but it was not associated with those diseases [25,26]. The promoter region variant, rs1122539, is associated with various diseases, including preeclampsia and arterial disease [13,27]. Moreover, the AA genotype of rs1122539 was significantly associated with increased MMP-8 serum levels and the A allele of rs1122539 was associated with an increased risk of arterial disease [27]. However, our data showed that only the intron variant (rs2509013 C>T), but not the promoter region variant (rs1122539 G>A), was associated with the decreased risk of idiopathic RPL. Therefore, our results suggest that rs1122539 does not affect RPL risk, even though rs1122539 may affect the levels of MMP-8 in the plasma or cervical fluid. 

MMP-27 is also known as epilysin. In terms of genomic location, *MMP-27* is located next to *MMP-8* on 11q22.2. *MMP-27* transcripts have been detected in almost all mouse tissues during development [5]. Additionally, MMP-27 is expressed in the human endometrium at the secretory phase [7]. However, no studies to date have examined the potential associations between polymorphisms in *MMP-27* and RPL risk. Our data suggest that functional dysregulation of *MMP-8* and *MMP-27* may contribute to early pregnancy failure either by affecting the ECM remodeling at the implantation site and/or by inhibiting effective endometrial or placental angiogenesis. However, further functional studies (in suitable animal models) exploring how polymorphisms in *MMP-8* and *MMP-27* may impact pregnancy outcomes, may shed light on such potential etiological mechanisms underlying RPL.

In our present study, the TC genotype of the *MMP-27* rs3809017 T>C polymorphism was associated with a decreased RPL risk. In addition, the levels of homocysteine and uric acid differed significantly between the genotypes of the *MMP-27* rs3809017 polymorphism in the patient group (Table 5). Interestingly, the TC type of the *MMP-27* rs3809017 T>C polymorphism, which was associated with decreased RPL risk in the genotype frequency analysis, showed the lowest levels of homocysteine and uric acid in the ANOVA analysis. Homocysteine and uric acid are important for pregnancy because elevated levels of homocysteine and uric acid can lead to various pregnancy complications, such as still birth, preterm birth, and pregnancy loss [28,29,30]. The differences in serum homocysteine and uric acid levels according to genotype were small; however, they may still be important for the process and outcomes of pregnancy. However, further functional studies examining the potential links between MMP-27 and the serum levels of homocysteine and uric acid are required. Moreover, we suggest that our findings should be replicated in a larger sample and using another population.

## 5. Conclusions

In conclusion, we investigated the potential associations between *MMP-8* rs2509013 C>T, *MMP-8* rs11225395 G>A, and *MMP-27* rs3809017 T>C polymorphisms and the risk of idiopathic RPL. The present study demonstrated, for the first time, that the TT genotype of *MMP-8* rs2509013 C>T and the TC genotype of *MMP-27* rs3809017 T>C were associated with a decreased RPL susceptibility. Additionally, this trend was maintained in the haplotype and genotype combination analyses. Moreover, the homocysteine and uric acid levels were significantly different according to the genotype of *MMP-27* rs3809017 T>C. Therefore, our study suggests a potential role for the *MMP-8* rs2509013 C>T and *MMP-27* rs3809017 T>C polymorphisms in RPL in Korean women. It is necessary to perform validation of the RPL risk-predictive utility of these polymorphisms in future large cohort retrospective and prospective studies that: (a) examine the associations between these polymorphisms and the serum levels of homocysteine and uric acid, and (b) connect these factors to the pathogenesis of RPL.

## Figures and Tables

**Table 1 genes-10-00347-t001:** Baseline clinical and laboratory profiles of the recurrent pregnancy loss (RPL) patients and control participants.

Characteristics	Controls (*n* = 240)	RPL Patients (*n* = 375)	*p*
Age (years)	33.19 ± 5.43	33.21 ± 4.60	0.965
BMI (kg/m^2^)	21.71 ± 3.45	21.5 ± 3.89	0.612
Previous pregnancy losses (n)	None	3.31 ± 1.83	
Live births (n)	1.75 ± 0.69	None	
Average gestational weeks	39.29 ± 1.64	7.33 ± 1.85	<0.0001
RPL < 10 weeks (%)	None	89.4	
CD56 NK Cells (%)	NA	18.25 ± 7.99	
Homocysteine (μmol/L)	NA	6.96 ± 2.04	
Folate (ng/mL)	NA	14.21 ± 12.00	
Total cholesterol (mg/dL)	NA	187.71 ± 49.89	
Urate (mg/dL)	NA	3.80 ± 0.84	
PAI-1 (ng/mL)	NA	10.52 ± 5.75	
PLT (10^3^/μL)	237.20 ± 66.69	254.00 ± 58.14	0.009
aPTT (sec)	33.44 ± 3.81	32.19 ± 4.19	0.034
PT	NA	11.58 ± 0.86	

RPL, recurrent pregnancy loss; BMI, body mass index; PAI-1, plasminogen activator inhibitor-1; PLT, platelet count; aPTT, activated partial thromboplastin time; PT, prothrombin time. *P* values indicate whether the proportional differences between the values of the parameter in the two groups were statistically significant.

**Table 2 genes-10-00347-t002:** Genotype frequencies of MMP gene polymorphisms in the RPL patients and control participants.

Genotypes	Controls (*n* = 240) (*n*, %)	PL ≥ 2 (*n* = 375) (*n*, %)	AOR (95% CI)	*p*	FDR-*P* ^a^	PL ≥ 3 (*n* = 138) (*n*, %)	AOR (95% CI)	*p*	FDR-*P* ^a^
***MMP-8* rs2509013 C>T**									
CC	74 (30.8)	144 (38.4)	1.000 (reference)			49 (35.5)	1.000 (reference)		
CT	109 (45.4)	185 (49.3)	0.867 (0.600–1.252)	0.447	0.671	73 (52.9)	1.015 (0.636–1.620)	0.951	0.951
TT	57 (23.8)	46 (12.3)	0.415 (0.257–0.671)	0.0003	0.001	16 (11.6)	0.424 (0.219–0.822)	0.011	0.033
Dominant (CC vs. CT + TT)			0.712 (0.504–1.004)	0.053	0.080		0.812 (0.521–1.266)	0.359	0.466
Recessive (CC + CT vs. TT)			0.450 (0.293–0.691)	0.0003	0.001		0.422 (0.232–0.770)	0.005	0.015
HWE-*P* ^b^	0.177	0.253				0.151			
***MMP-8* rs11225395 G>A**									
GG	182 (75.8)	290 (77.3)	1.000 (reference)			113 (81.9)	1.000 (reference)		
GA	54 (22.5)	79 (21.1)	0.922 (0.622–1.365)	0.683	0.683	23 (16.7)	0.683 (0.397–1.174)	0.168	0.504
AA	4 (1.7)	6 (1.6)	0.950 (0.264–3.415)	0.937	0.937	2 (1.4)	0.845 (0.152–4.705)	0.847	0.847
Dominant (GG vs. GA + AA)			0.923 (0.630–1.352)	0.680	0.680		0.693 (0.410–1.172)	0.171	0.466
Recessive (GG + GA vs. AA)			0.963 (0.269–3.451)	0.954	0.954		0.906 (0.163–5.027)	0.910	0.910
HWE-*P* ^b^	0.998	0.816				0.512			
***MMP-27* rs3809017 T>C**									
TT	98 (40.8)	186 (49.6)	1.000 (reference)			61 (44.2)	1.000 (reference)		
TC	116 (48.3)	151 (40.3)	0.681 (0.483–0.961)	0.029	0.087	63 (45.7)	0.857 (0.549–1.338)	0.497	0.746
CC	26 (10.8)	38 (10.1)	0.751 (0.429–1.315)	0.316	0.474	14 (10.1)	0.838 (0.402–1.745)	0.636	0.847
Dominant (TT vs. TC + CC)			0.697 (0.502–0.968)	0.031	0.080		0.854 (0.558–1.307)	0.466	0.466
Recessive (TT + TC vs. CC)			0.929 (0.547–1.579)	0.786	0.954		0.894 (0.447–1.788)	0.752	0.910
HWE-*p* ^b^	0.335	0.372				0.700			

MMP, matrix metalloproteinase; RPL, recurrent pregnancy loss; PL, pregnancy loss; AOR, adjusted odds ratio; CI, confidence interval. AOR was adjusted by the age of participants. ^a^ False discovery rate *P*-value. ^b^ Hardy-Weinberg Equilibrium *p*-value.

**Table 3 genes-10-00347-t003:** Haplotype analysis of *MMP* gene polymorphisms in the RPL patients and control participants.

Haplotypes	Control (*2n* = 480) (*n*, %)	RPL (*2n* = 750) (*n*, %)	OR (95% CI)	*p* ^a^	*FDR-P* ^b^
***MMP-8* rs2509013 C>T/*MMP-8* rs11225395 G>A/*MMP-27* rs3809017 T>C**					
C-G-T	160 (33.4)	402 (53.5)	1.000 (reference)		
C-G-C	87 (18.1)	39 (5.2)	0.178 (0.117–0.272)	<0.0001	0.0004
C-A-T	7 (1.4)	33 (4.4)	1.876 (0.813–4.329)	0.148	0.173
C-A-C	3 (0.7)	0 (0.0)	0.057 (0.003–1.110)	0.024	0.034
T-G-T	109 (22.8)	47 (6.2)	0.172 (0.116–0.253)	<0.0001	0.0004
T-G-C	61 (12.8)	172 (22.9)	1.122 (0.795–1.584)	0.543	0.543
T-A-T	36 (7.4)	42 (5.6)	0.464 (0.287–0.752)	0.002	0.005
T-A-C	17 (3.5)	16 (2.2)	0.375 (0.185–0.760)	0.010	0.018
***MMP-8* rs2509013 C>T/*MMP-8* rs11225395 G>A**					
C-G	247 (51.5)	439 (58.6)	1.000 (reference)		
C-A	10 (2.1)	34 (4.5)	1.913 (0.929–3.939)	0.102	0.102
T-G	171 (35.6)	220 (29.3)	0.724 (0.562–0.933)	0.014	0.038
T-A	52 (10.9)	57 (7.6)	0.617 (0.411–0.927)	0.025	0.038
***MMP-8* rs2509013 C>T/*MMP-27* rs3809017 T>C**					
C-T	167 (34.8)	434 (57.9)	1.000 (reference)		
C-C	90 (18.8)	39 (5.2)	0.167 (0.110–0.253)	<0.0001	0.0002
T-T	145 (30.2)	89 (11.9)	0.236 (0.172–0.325)	<0.0001	0.0002
T-C	78 (16.2)	188 (25.1)	0.927 (0.674–1.276)	0.683	0.683
***MMP-8* rs11225395 G>A/*MMP-27* rs3809017 T>C**					
G-T	270 (56.2)	447 (59.7)	1.000 (reference)		
G-C	148 (30.9)	212 (28.2)	0.865 (0.668–1.121)	0.289	0.434
A-T	42 (8.8)	76 (10.1)	1.093 (0.728–1.641)	0.683	0.683
A-C	20 (4.1)	15 (2.1)	0.453 (0.228–0.900)	0.031	0.093

MMP, matrix metalloproteinase; RPL, recurrent pregnancy loss; OR, odds ratio; CI, confidence interval. ^a^ Fisher’s exact test. ^b^ False discovery rate *p*-value.

**Table 4 genes-10-00347-t004:** Genotype combination analysis of *MMP* gene polymorphisms between the RPL patients and control participants.

Genotype Combinations	Control (N = 240) (*n*, %)	RPL (N = 375) (*n*, %)	AOR (95% CI)	*p*	*FDR-P* ^a^
***MMP-8* rs2509013/*MMP-8* rs11225395**					
CC/GG	70 (29.2)	126 (33.6)	1.000 (reference)		
CC/GA	4 (1.7)	16 (4.3)	2.231 (0.717–6.941)	0.166	0.387
CC/AA	0 (0.0)	2 (0.5)	NA	NA	NA
CT/GG	81 (33.8)	135 (36.0)	0.924 (0.617–1.382)	0.699	0.816
CT/GA	25 (10.4)	47 (12.5)	1.044 (0.593–1.839)	0.882	0.882
CT/AA	3 (1.3)	3 (0.8)	0.579 (0.113–2.981)	0.513	0.816
TT/GG	31 (12.9)	29 (7.7)	0.519 (0.289–0.933)	0.028	0.098
TT/GA	25 (10.4)	16 (4.3)	0.354 (0.177–0.708)	0.003	0.021
TT/AA	1 (0.4)	1 (0.3)	0.556 (0.034–9.027)	0.680	0.816
***MMP-8* rs2509013/*MMP-27* rs3809017**					
CC/TT	32 (13.3)	123 (32.8)	1.000 (reference)		
CC/TC	32 (13.3)	17 (4.5)	0.137 (0.068–0.279)	<0.0001	0.0003
CC/CC	10 (4.2)	4 (1.1)	0.102 (0.030–0.351)	0.0003	0.0005
CT/TT	42 (17.5)	56 (14.9)	0.347 (0.198–0.609)	0.0002	0.0004
CT/TC	58 (24.2)	120 (32.0)	0.530 (0.321–0.875)	0.013	0.015
CT/CC	9 (3.8)	9 (2.4)	0.260 (0.095–0.709)	0.009	0.012
TT/TT	24 (10.0)	7 (1.9)	0.077 (0.030–0.196)	<0.0001	0.0003
TT/TC	26 (10.8)	14 (3.7)	0.137 (0.064–0.294)	<0.0001	0.0003
TT/CC	7 (2.9)	25 (6.7)	0.953 (0.376–2.413)	0.919	0.919
***MMP-8* rs11225395/*MMP-27* rs3809017**					
GG/TT	76 (31.7)	136 (36.3)	1.000 (reference)		
GG/TC	85 (35.4)	121 (32.3)	0.789 (0.532–1.171)	0.240	0.578
GG/CC	21 (8.8)	33 (8.8)	0.869 (0.469–1.612)	0.656	0.869
GA/TT	19 (7.9)	46 (12.3)	1.351 (0.738–2.472)	0.330	0.578
GA/TC	30 (12.5)	28 (7.5)	0.520 (0.289–0.936)	0.029	0.203
GA/CC	5 (2.1)	5 (1.3)	0.490 (0.135–1.782)	0.279	0.578
AA/TT	3 (1.3)	4 (1.1)	0.776 (0.169–3.574)	0.745	0.869
AA/TC	1 (0.4)	2 (0.5)	1.145 (0.102–12.86)	0.912	0.912

MMP, matrix metalloproteinase; RPL, recurrent pregnancy loss; AOR, adjusted odds ratio; CI, confidence interval ^a^ False discovery rate *p*-value.

**Table 5 genes-10-00347-t005:** Differences in the clinical parameters according to genotype of MMP gene polymorphisms in the RPL patient group.

Genotypes	Hcy (μmol/L)	Folate (ng/mL)	PLT (10^3^/μL)	NK cell (%)	PT (sec)	Uric acid (mg/dL)	T. chol (mg/dL)	PAI-1 (mg/dL)
	Mean ± SD	Mean ± SD	Mean ± SD	Mean ± SD	Mean ± SD	Mean ± SD	Mean ± SD	Mean ± SD
***MMP-8* rs2509013 C>T**								
CC	7.28 ± 2.49	12.67 ± 7.51	249.35 ± 53.34	18.46 ± 7.22	11.59 ± 0.93	3.88 ± 0.85	195.63 ± 52.72	10.83 ± 5.83
CT	6.77 ± 1.56	15.34 ± 14.88	254.54 ± 56.18	17.55 ± 8.74	11.59 ± 0.81	3.73 ± 0.85	180.79 ± 49.16	9.95 ± 5.60
TT	6.71 ± 2.01	14.14 ± 8.89	266.80 ± 77.93	20.32 ± 7.00	11.48 ± 0.88	3.80 ± 0.79	190.52 ± 41.64	12.06 ± 6.22
*p*	0.306 ^a^	0.317 ^b^	0.423 ^b^	0.418 ^b^	0.849 ^b^	0.571 ^b^	0.182 ^b^	0.369 ^b^
***MMP-8* rs11225395 G>A**								
GG	6.92 ± 2.05	14.09 ± 12.32	249.05 ± 50.24	18.17 ± 8.15	11.64 ± 0.87	3.80 ± 0.82	186.77 ± 51.91	10.52 ± 5.72
GA	7.14 ± 2.10	15.45 ± 10.90	269.84 ± 78.50	19.30 ± 6.90	11.33 ± 0.80	3.85 ± 0.92	192.47 ± 43.87	10.97 ± 6.47
AA	7.15 ± 0.68	9.66 ± 7.65	274.50 ± 70.90	16.08 ± 9.18	11.40 ± 0.71	2.93 ± 0.55	173.67 ± 10.26	8.55 ± 3.40
*p*	0.762 ^b^	0.574 ^b^	0.216 ^a^	0.705 ^b^	0.124 ^b^	0.191 ^b^	0.738 ^b^	0.753 ^b^
***MMP-27* rs3809017 T>C**								
TT	7.27 ± 2.31	13.13 ± 7.77	252.43 ± 55.94	17.81 ± 7.38	11.51 ± 0.92	3.95 ± 0.88	188.89 ± 50.04	10.22 ± 5.96
TC	6.63 ± 1.63	14.99 ± 15.59	254.50 ± 57.40	18.30 ± 8.69	11.70 ± 0.79	3.62 ± 0.80	182.71 ± 47.63	10.86 ± 5.49
CC	6.85 ± 1.98	15.76 ± 9.07	260.35 ± 73.49	20.15 ± 7.05	11.43 ± 0.82	3.68 ± 0.65	199.42 ± 56.91	10.53 ± 6.19
*p*	0.042 ^b^	0.462 ^b^	0.853 ^b^	0.655 ^b^	0.225 ^b^	0.046 ^b^	0.417 ^b^	0.842 ^b^

MMP, matrix metalloproteinase; RPL, recurrent pregnancy loss; Hcy, homocysteine; PLT, platelet count; NK cell, natural killer cell; PT, prothrombin time; PAI-1, plasminogen activator inhibitor-1. ^a^ Calculated using the Kruskal–Wallis test. ^b^ Calculated using ANOVA.

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
