# Peer review of "Association Study between the Polymorphisms of Matrix Metalloproteinase (MMP) Genes and Idiopathic Recurrent Pregnancy Loss"

_genes, 2019, doi:10.3390/genes10050347_

Round 1

Reviewer 1 Report

-Study population (Korean women) should be mentioned in the abstract

-sample sizes are swapped between abstract (240 cases an 375 controls) and Methods section 2.1 (375 patients and 240 controls)

-line 104 “The women in the control group were included..” should remove “were” -> “The women in the control group included..”

-Where are MMP-8 and MMP-27 located in the genome? Are they nearby? If not, it does not make sense to perform haplotype analysis including all three SNPs. (give chromosomal locations and if possible LD between these SNPs in the Korean population). I eventually found the chromosomal locations in the discussion-- can this information be provided earlier to give context for the haplotype analyses?

-small sample size should be added to the limitations of this study

-have these specific polymorphisms been investigated in RPL in other populations?

Author Response

Cover Letter

Editor and Reviewer

Genes

April, 27, 2019

Thank you for reviewing our manuscript. The manuscript has changed according to reviewer’s suggestion. The changed or added sentences were marked with track changed function in Microsoft word, and the related reviewer’s comments and our responses were added in memo. Additionally, we have used the English editing service (BioScienceWriters LLC, Houston TX) before first submission the manuscript and we have check one more before revision submission. But the manuscript may have some grammatical errors. Please, let us know if there is a grammatical error in the manuscript.

Kind regards,

Prof. Nam Keun Kim, PhD

Department of Biomedical Science,

College of Life Science,

CHA University.

CHA Bio Complex,

335 Pangyo-ro, Bundang-gu,

Seongnam 13488, South Korea

TEL : +82-31-881-7137, +82-10-4322-7515

FAX: +82-31-881-7249

E-mail : nkkim@cha.ac.kr, namkkim@naver.com

Reviewer 1.

-Study population (Korean women) should be mentioned in the abstract

- Author’s response: Thank you for your valuable time for review. ‘Korean’ was added in methods and material of abstract.

-sample sizes are swapped between abstract (240 cases an 375 controls) and Methods section 2.1 (375 patients and 240 controls)

- Author’s response: Thank you for your comment. The sample sizes in Methods section are correct. The sample sizes in abstract have changed.

-line 104 “The women in the control group were included..” should remove “were” -> “The women in the control group included..”

 - Author’s response: Thank you for your comment. The sentence has changed.

-Where are MMP-8 and MMP-27 located in the genome? Are they nearby? If not, it does not make sense to perform haplotype analysis including all three SNPs. (give chromosomal locations and if possible LD between these SNPs in the Korean population). I eventually found the chromosomal locations in the discussion-- can this information be provided earlier to give context for the haplotype analyses?

- Author’s response: Thank you for providing these insights. We already have made a LD block using those three SNPs, but it was not interesting data. So, we just briefly added that those two genes are on the same chromosome at ‘3.3 Haplotype analysis’ in ‘result section’.

-small sample size should be added to the limitations of this study

- Author’s response: Thank you for your suggestion. The mention about limitation of sample size was added at last paragraph in discussion section.

-have these specific polymorphisms been investigated in RPL in other populations?

- Author’s response: To our knowledge, these polymorphisms have not been investigated in RPL in other populations.

Reviewer 2 Report

This study purports that two new polymorphisms, an intronic variant of MMP8 and MMP-27, are associated with the risk of recurrent pregnancy loss (RPL). Overall the study is well written and appears technically sound.  However, the design of the study is open to criticisms and a number of issues need addressing:

1. Although the authors provide a plausible narrative to implicate a role for MMP-8 and MMP-27 in RPL, how these particular polymorphisms affect the expression or function of these metalloproteinases is unclear and no attempt is made to address this issue experimentally. Furthermore, the authors do explain the mechanistic link, if any, between circulating homocysteine/uric acid levels and MMP27 genotype.

2. In excess of 100 susceptibility genes have now been linked to RPL (doi.org/10.1016/j.bpobgyn.2014.12.001). However, in most cases, the clinical significance of these polymorphisms on pregnancy outcome in RPL patients has not been tested in prospective cohort studies. The few polymorphisms that have been tested were found not to impact subsequent pregnancy outcome. In this respect, this report just adds to the already substantial list of polymorphisms claimed to be associated with RPL without any evidence of clinical relevance.

3. The authors studied patients with ‘idiopathic/unexplained ’ RPL. It is indeed widely assumed that a range of subclinical disorders, e.g. endocrine perturbations, uterine anomalies, latent infections, thrombophilias, immune disorders, etc., are causal to RPL. However, the assumption that subclinical disorders are causal is just not supported by current clinical evidence (see latest ESHRE Guidelines: https://www.eshre.eu/Guidelines-and-Legal/Guidelines/Recurrent-pregnancy-loss.aspx) or by evidence that emerged from interventional randomized clinical trials. If the authors wish to adhere to the widely-promulgated but misleading concept of ‘explained’ and ‘unexplained’ RPL, they should have determined the incidence of MMP8 and MMP27 polymorphisms in women with ‘explained’ RPL.

4. RPL is not a uniform syndrome nor precludes a future successful pregnancy. In fact, the cumulative live birth rate is high in RPL (>65% after 3 consecutive miscarriages), although the likelihood of a future successful pregnancy gradually decreases. So was there evidence that the polymorphisms were less/more prevalent in women with higher-order miscarriages? Further, RPL due to bleeding in early pregnancy is unlikely to share the same etiology as consecutive missed miscarriages around 12 weeks of pregnancy. This study, like many others, lacks this level of clinical granularity and just lump all miscarriages together.

Author Response

Cover Letter

Editor and Reviewer

Genes

April, 27, 2019

Thank you for reviewing our manuscript. The manuscript has changed according to reviewer’s suggestion. The changed or added sentences were marked with track changed function in Microsoft word, and the related reviewer’s comments and our responses were added in memo. Additionally, we have used the English editing service (BioScienceWriters LLC, Houston TX) before first submission the manuscript and we have check one more before revision submission. But the manuscript may have some grammatical errors. Please, let us know if there is a grammatical error in the manuscript.

Kind regards,

Prof. Nam Keun Kim, PhD

Department of Biomedical Science,

College of Life Science,

CHA University.

CHA Bio Complex,

335 Pangyo-ro, Bundang-gu,

Seongnam 13488, South Korea

TEL : +82-31-881-7137, +82-10-4322-7515

FAX: +82-31-881-7249

E-mail : nkkim@cha.ac.kr, namkkim@naver.com

Reviewer 2.

This study purports that two new polymorphisms, an intronic variant of MMP8 and MMP-27, are associated with the risk of recurrent pregnancy loss (RPL). Overall the study is well written and appears technically sound.  However, the design of the study is open to criticisms and a number of issues need addressing:

- Author’s response: Thank you for your valuable time to review our paper. We have tried our best to answer your queries.

1. Although the authors provide a plausible narrative to implicate a role for MMP-8 and MMP-27 in RPL, how these particular polymorphisms affect the expression or function of these metalloproteinases is unclear and no attempt is made to address this issue experimentally. Furthermore, the authors do explain the mechanistic link, if any, between circulating homocysteine/uric acid levels and MMP27 genotype.

 - Author’s response: Thank you for your suggestion. We agree with your opinion. We didn’t try to identify the role of those two SNPs on intron region in this study. However, this is very important question about relation of SNPs, MMP proteins, and RPL susceptibility. Furthermore, the homocysteine and uric acid were reported that induce and decrease MMP-9, respectively. Therefore, the levels of homocysteine and uric acid according to genotypes of MMP-27 were very interesting because they may be suggested as a new MMP signaling pathway or interaction with MMP-9. So, we are preparing that try to the functional studies of SNP and investigation of relation between MMP-27 and levels of homocysteine/uric acid.

2. In excess of 100 susceptibility genes have now been linked to RPL (doi.org/10.1016/j.bpobgyn.2014.12.001). However, in most cases, the clinical significance of these polymorphisms on pregnancy outcome in RPL patients has not been tested in prospective cohort studies. The few polymorphisms that have been tested were found not to impact subsequent pregnancy outcome. In this respect, this report just adds to the already substantial list of polymorphisms claimed to be associated with RPL without any evidence of clinical relevance.

- Author’s response: We agree with your opinion that too much SNP associated with RPL are reported. However, we think that the study for RPL associated SNP is important for this reason. The various reports about RPL including SNP studies were published every year. But, the RPL still has unknown reason. Therefore, we think that the unknown reason include very complex genetic issue, not problem with single gene and we will study about RPL associated SNP till the reasons of RPL were clarified.

3. The authors studied patients with ‘idiopathic/unexplained ’ RPL. It is indeed widely assumed that a range of subclinical disorders, e.g. endocrine perturbations, uterine anomalies, latent infections, thrombophilias, immune disorders, etc., are causal to RPL. However, the assumption that subclinical disorders are causal is just not supported by current clinical evidence (see latest ESHRE Guidelines: https://www.eshre.eu/Guidelines-and-Legal/Guidelines/Recurrent-pregnancy-loss.aspx) or by evidence that emerged from interventional randomized clinical trials. If the authors wish to adhere to the widely-promulgated but misleading concept of ‘explained’ and ‘unexplained’ RPL, they should have determined the incidence of MMP8 and MMP27 polymorphisms in women with ‘explained’ RPL.

- Author’s response: Thank you for providing these insights. We agree with you. So, the all participants didn’t have subclinical disorders in this study. Moreover, the patients were classified as ‘idiopathic’ because they were very healthy and had no harmful habit for pregnancy such as smoking or drinking. However, we didn’t think about the ‘explained’ RPL. The approach for case-control study was very interesting. But, we don’t have the samples of ‘explained’ RPL patients. Therefore, if we got the chance for the collect the samples of ‘explained’ RPL patients, we will try to new concept of case-control study as you said.

4. RPL is not a uniform syndrome nor precludes a future successful pregnancy. In fact, the cumulative live birth rate is high in RPL (>65% after 3 consecutive miscarriages), although the likelihood of a future successful pregnancy gradually decreases. So was there evidence that the polymorphisms were less/more prevalent in women with higher-order miscarriages? Further, RPL due to bleeding in early pregnancy is unlikely to share the same etiology as consecutive missed miscarriages around 12 weeks of pregnancy. This study, like many others, lacks this level of clinical granularity and just lump all miscarriages together.

- Author’s response: Thank you for your suggestion. We agree with you and we also know about the successful live birth in future pregnancy of RPL patients. However, some RPL patients are still difficult to have a baby, although they had no specific reason. Moreover, the live birth of those RPL patients will more difficult with time. Therefore, we try to found the associated SNP with RPL susceptibility before investigate a genetic effect for the severity of RPL. Additionally, we already have DNA samples of RPL patients with six or more pregnancy loss (most pregnancy lose number is 14) although, the sample cannot used for make individual RPL subgroups for statistical analysis because the sample size is too small. We will endeavor to investigate the association between those SNP and RPL severity if we collect enough samples of RPL patients with high number of pregnancy loss for make individual RPL subgroups. Additionally, RPL patients that had bleeding issue and taken hormone to prevent pregnancy loss were excluded in participants of this study.